# WGAN-GP for Synthetic Retinal Image Generation: Enhancing Sensor-Based Medical Imaging for Classification Models

**DOI:** 10.3390/s25010167

**Published:** 2024-12-31

**Authors:** Héctor Anaya-Sánchez, Leopoldo Altamirano-Robles, Raquel Díaz-Hernández, Saúl Zapotecas-Martínez

**Affiliations:** 1Computer Science Department, Instituto Nacional de Astrofísica Óptica y Electrónica, Luis Enrrique Erro No. 1, Sta. María Tonantzintla, Puebla 72840, Mexico; hector.anaya@inaoep.mx (H.A.-S.); szapotecas@inaoep.mx (S.Z.-M.); 2 Optics Department, Instituto Nacional de Astrofísica Óptica y Electrónica, Luis Enrrique Erro No. 1, Sta. María Tonantzintla, Puebla 72840, Mexico; raqueld@inaoep.mx

**Keywords:** generative AI, GANs, AI, machine learning, retinal images

## Abstract

Accurate synthetic image generation is crucial for addressing data scarcity challenges in medical image classification tasks, particularly in sensor-derived medical imaging. In this work, we propose a novel method using a Wasserstein Generative Adversarial Network with Gradient Penalty (WGAN-GP) and nearest-neighbor interpolation to generate high-quality synthetic images for diabetic retinopathy classification. Our approach enhances training datasets by generating realistic retinal images that retain critical pathological features. We evaluated the method across multiple retinal image datasets, including Retinal-Lesions, Fine-Grained Annotated Diabetic Retinopathy (FGADR), Indian Diabetic Retinopathy Image Dataset (IDRiD), and the Kaggle Diabetic Retinopathy dataset. The proposed method outperformed traditional generative models, such as conditional GANs and PathoGAN, achieving the best performance on key metrics: a Fréchet Inception Distance (FID) of 15.21, a Mean Squared Error (MSE) of 0.002025, and a Structural Similarity Index (SSIM) of 0.89 in the Kaggle dataset. Additionally, expert evaluations revealed that only 56.66% of synthetic images could be distinguished from real ones, demonstrating the high fidelity and clinical relevance of the generated data. These results highlight the effectiveness of our approach in improving medical image classification by generating realistic and diverse synthetic datasets.

## 1. Introduction

The accurate classification of medical images is crucial for the effective diagnosis and treatment of numerous diseases. Nevertheless, medical image classification systems frequently encounter obstacles, mainly when dealing with insufficient and imbalanced datasets or unlabeled datasets, as annotating retinal images requires specialized expertise and is time-consuming and costly [1,2], which can significantly affect their accuracy and overall performance. Additionally, retinal images are prone to variations in capture conditions, such as differences in lighting, focus, and imaging equipment, which can introduce noise and inconsistencies, leading to poor model generalization and overfitting [3]. Traditional data augmentation techniques, such as geometric transformations (e.g., rotation, translation, zoom), are commonly employed to mitigate these dataset imbalances and are often insufficient for generating the subtle pathological variations needed to improve model performance [4]. Furthermore, the detection of fine-grained pathological features, including microaneurysms, hemorrhages, and exudates, remains challenging due to their small size and variability [5]. These limitations highlight the need for more sophisticated data augmentation methods that can generate realistic and diverse synthetic images while preserving critical diagnostic features. With the advent of artificial intelligence and deep learning, more sophisticated methods have emerged. One such method is Generative Adversarial Networks (GAN) [6], which are capable of generating realistic synthetic images that are virtually indistinguishable from real images. Despite this, traditional GANs often suffer from issues such as mode collapse, which limits the diversity of generated images [7]. This has led to the exploration of more advanced augmentation methods using GANs. For instance, recent studies such as the AC-WGAN-GP framework proposed by Sun et al. [8] have demonstrated the efficacy of GANs in generating high-quality synthetic samples for hyperspectral image classification. This framework highlights the importance of integrating techniques like auxiliary classifiers and gradient penalties to ensure stable training and high-quality sample generation. These principles are equally applicable in medical image analysis, where generating high-fidelity images is crucial for improving classification performance. Our proposed method using Wasserstein GAN with Gradient Penalty (WGAN-GP) aims to overcome these challenges by ensuring stable training and generating medically plausible synthetic images, thereby addressing the data scarcity problem and enhancing the performance of retinal image classification models.

GANs have become increasingly important in the field of medical image processing, where they are widely used to create synthetic datasets that help improve classification performance across various medical tasks. For instance, ref. [9] demonstrated the effectiveness of using GAN-augmented datasets to enhance diagnostic accuracy in the classification of liver lesions. Their application of a Deep Convolutional GAN (DCGAN) [10] notably increased dataset diversity, leading to a marked improvement in model performance.

Recent advancements have continued to push the boundaries of synthetic image generation in medical image analysis. For example, ref. [11] introduced a novel architecture utilizing conditional GANs (cGANs) [12] to generate realistic mammography images containing synthetic lesions. By allowing for controlled variation in lesion types and their specific positions, their approach tackled the challenge of limited data availability in breast cancer screening. This technique led to significant improvements in the performance of classification models trained on their augmented datasets, providing a promising solution for enhancing breast cancer diagnosis.

In a related vein, ref. [13] proposed an explainable GAN framework designed to synthesize retinal images for the detection of diabetic retinopathy. Their work underscored the critical need for interpretability in GAN models, ensuring that medical professionals can understand and trust the synthetic images used in training classification systems. The incorporation of attention mechanisms within their framework further enhanced the generation of lesions, ensuring that the synthesized images were closely aligned with clinical standards and expectations. This focus on explainability adds a layer of transparency, which is essential in medical applications.

Similarly, ref. [14] introduced a multiscale GAN architecture aimed at generating high-resolution medical images, with a particular focus on improving the granularity of synthetic data. This work is especially relevant for generating small-scale lesions, which are critical for early diagnosis of various medical conditions. The multiscale approach effectively addressed the challenge of generating fine details in synthetic images, overcoming a limitation that had hindered previous methods in capturing the necessary resolution for accurate diagnosis.

Building on these advancements, we propose the use of a conditional Generative Adversarial Network (cGAN) for generating synthetic medical images tailored for classification tasks. Specifically, we adopt a Wasserstein GAN with Gradient Penalty (WGAN), ref. [15] framework to ensure the production of high-quality images while mitigating the risk of mode collapse, a frequent issue in GAN-based models. Our approach integrates medical domain knowledge directly into the image generation process, ensuring that the resulting synthetic images are not only visually convincing but also medically plausible, aligning with the diagnostic requirements of healthcare professionals.

The main contributions of this work are as follows:Application of WGAN-GP for Enhanced Stability: The use of WGAN-GP ensures the generation of high-quality images while avoiding issues such as mode collapse, which is a common problem in traditional GANs. This enhancement contributes to more stable training and results in superior image synthesis, making the model more reliable for medical applications.Lesion Extraction and Style Transfer: We ensure that the synthetic images retain critical pathological features by incorporating lesion extraction and style transfer techniques into the GAN model. This incorporation makes the generated data particularly valuable for training classification models in medical imaging, as it closely mirrors real-world medical conditions.Dataset Augmentation for Improved Model Performance: The synthetic images produced by our method are used to augment real datasets, addressing the challenge of data scarcity in medical imaging. This augmentation leads to improved performance in medical image classification models, as the expanded dataset allows the model to learn from a more diverse set of examples.

Building on the need for image augmentation in medical contexts, this work presents a novel approach using WGAN-GP to generate high-quality synthetic retinal images. The subsequent sections provide a detailed breakdown of our methodology and experimental design. In Section 2, we describe the core components of the proposed method, including the application of WGAN-GP, lesion extraction, style transfer, and preprocessing steps tailored for medical image synthesis. Section 3 outlines the datasets used and the specific configurations for each, followed by Section 4, which analyzes our experimental results, comparing the generated images across various metrics and databases. The paper concludes with a discussion of key findings, expert evaluations, and areas for future improvement in synthetic medical image generation.

## 2. Materials and Metods

Our proposed method leverages WGAN-GP to generate synthetic medical images with the goal of augmenting datasets used for classification tasks. The framework, illustrated in Figure 1, comprises several of several key components, each designed to ensure high-quality image generation and robust model training:

### 2.1. WGAN-GP

In our proposed method, the Wasserstein Generative Adversarial Network with Gradient Penalty (WGAN-GP) [15] was specifically chosen to improve the quality and stability of synthetic medical image generation. WGAN-GP addresses issues such as mode collapse, which commonly affects traditional GANs, by employing a Wasserstein loss and incorporating a gradient penalty to enforce the Lipschitz constraint during training. This method ensures smoother gradients and leads to more stable training, which is crucial for generating high-quality medical images.

In recent years, more advanced methods have emerged for image generation. StyleGAN and StyleGAN2 [16,17] introduced a style-based architecture and progressive growing to achieve high-resolution image synthesis with precise control over visual attributes. While StyleGAN excels at generating natural images, it requires large datasets and significant computational resources, which limits its applicability in medical imaging. In contrast, WGAN-GP remains robust with smaller datasets and simpler architectures.

BigGAN [18] enhances image quality by scaling up model size and dataset scope, using class-conditional generation and orthogonal regularization. However, BigGAN’s reliance on extensive computational power and large-scale datasets makes it less practical for medical image synthesis. WGAN-GP offers a more efficient alternative while maintaining high fidelity.

Diffusion models, such as Denoising Diffusion Probabilistic Models (DDPMs) [19,20], have gained attention for their ability to produce high-fidelity images through an iterative denoising process. Despite their impressive results, diffusion models require high computational costs and slow inference times, making them less suitable for rapid image generation tasks. In comparison, WGAN-GP’s adversarial framework offers faster training and inference.

Methods integrating multimodal techniques have also advanced image generation capabilities. MEGAN (Mixture of Experts of Generative Adversarial Networks) [21] leverages a mixture of expert GANs to generate images conditioned on multiple modalities, such as text, images, or categorical data. By combining outputs from specialized GANs, MEGAN achieves higher diversity and quality in multimodal image synthesis. These techniques can enhance medical image generation by incorporating additional modalities, such as clinical reports or annotations, alongside image data. However, these methods often require more complex architectures and training strategies, which may not always be feasible in data-limited medical contexts.

Generative Adversarial MultiTask Learning [22] generates images while simultaneously performing auxiliary tasks (e.g., face recognition and sketch synthesis). This approach improves the quality of the generated images by leveraging multiple related tasks during training. In medical imaging, similar multitask methods can enhance synthesis by incorporating diagnostic tasks, ensuring the generated images are clinically relevant and useful for downstream applications. While multitask GANs offer these advantages, their training can be more challenging due to the need for balanced learning across multiple objectives.

GANs with Transformers (GANformer) [23] combine GANs with transformer-based attention mechanisms to model long-range dependencies in images, improving global coherence. Latent Diffusion Models (LDMs) [24] optimize the diffusion process by operating in a lower-dimensional latent space, reducing computational demands while maintaining high-resolution outputs. Though powerful, these models introduce additional complexity and dependencies that may not always align with medical image generation requirements.

Compared with these newer methods, WGAN-GP remains a compelling choice due to its stability, efficiency, and ability to perform well on smaller datasets. This makes it particularly effective for generating synthetic medical images where preserving pathological details, computational feasibility, and robustness are paramount.

In our implementation, the generator network $G_{\theta }$ takes a vessel segmentation image $y\in {\left\{0,1\right\}}^{W\times H}$, lesion descriptors D(x), and a noise vector $z\in R^{Z}$ as inputs. Here, *y* and D(x) provide physiological and pathological information, respectively. The noise vector *z* introduces randomness, allowing the model to produce different outputs each time. The output is a synthesized diabetic retinopathy fundus image $\hat{x}\in {\left[0,1\right]}^{W\times H\times 3}$.

To ensure the stability of the training process and enhance the quality of the synthesized images, we employ the WGAN-GP. This approach optimizes the generator to produce high-quality images while maintaining smooth gradient behavior, thereby avoiding common issues like mode collapse. The entire synthesis process can be expressed as $\hat{x}=G_{\theta }\left(y,D\left(x\right),z\right)$, capturing the interplay of input features and randomness in the generation of realistic medical images.

We use the WGAN-GP strategy by solving the following optimization problem, which governs the interaction between the generator $G_{\theta }$ and the discriminator $D_{\gamma }$:(1)$$\begin{array}{ccc} \min_{\theta }\max_{\gamma }L\left(G_{\theta },D_{\gamma }\right) & = & E_{x,y}\left[L_{W}\left(x,\hat{x},y;\theta ,\gamma \right)+w_{p}\cdot L_{\mathrm{percept}}\left(x,\hat{x};\theta \right)\right.\\ & & \left.+w_{s}\cdot L_{\mathrm{severity}}\left(x,\hat{x};\theta \right)\right] \end{array}$$
where $L_{W}$ represents the Wasserstein loss, with $L_{\mathrm{percept}}$ and $L_{\mathrm{severity}}$ being the perceptual loss and severity loss, respectively.

Wasserstein Loss: Wasserstein loss is computed on real and synthesized images, where $L_{W}$ is defined as:(2)$$\begin{array}{c} L_{W}=D_{\gamma }\left(x,y\right)-D_{\gamma }\left(G_{\theta }\left(y,z\right),y\right) \end{array}$$

In this case, learning the discriminator parameter γ involves maximizing the Wasserstein loss $L_{W}$, which ensures the discriminator provides meaningful feedback during training. To maintain stable training and avoid gradient explosions or vanishing gradients, a gradient penalty is applied, promoting smooth updates to the model parameters.

For the generator, learning the parameter θ corresponds to minimizing the following loss function:(3)$$\begin{array}{c} L_{G}=-D_{\gamma }\left(G_{\theta }\left(y,z\right),y\right)+w_{p}\cdot L_{\mathrm{percept}}+w_{s}\cdot L_{\mathrm{severity}} \end{array}$$

This objective encourages the generator to produce synthetic images that the discriminator finds indistinguishable from real ones, thereby improving the overall quality of the generated data. The interplay between these two loss functions drives the adversarial training process, ultimately leading to the generation of realistic and clinically relevant medical images.

### 2.2. Lesion Extraction

Inspired by the work of [13], we employed the DR Detector from Kaggle’s Diabetic Retinopathy Detection competition [25], where the objective was to detect Diabetic Retinopathy (DR) and classify it into five severity levels: 0 (No DR), 1, 2, 3, and 4 according to the International Clinical Diabetic Retinopathy Disease Severity Scale [26]. The second-place team (Team ‘o_O’) publicly shared their implementation, which we utilized as the foundation for our lesion extraction process.

Using this implementation, we calculated the backpropagation of the lesion activations generated by the network. To refine the extracted lesions, we applied a Gaussian filter followed by binarization, resulting in two tensors representing lesions of different sizes. This approach, shown in Figure 2, allowed us to isolate and capture the critical pathological features in the retinal images, which are crucial for further analysis.

### 2.3. Style Transfer

To ensure our generated images closely resemble real images, we adopted a style transfer [27] technique that employs two distinct loss functions as shown in Figure 3. The first, called severity loss, leverages the DR Detector from the previous section (Team ‘o_O’ detector) to evaluate the grading of real and synthetic images. The severity loss is calculated as follows:(4)$$\begin{array}{c} L_{severity}=||DR\left(x\right)-DR\left(\hat{x}\right)\left|\right| \end{array}$$

This loss function measures the difference between the diabetic retinopathy grading of the real image *x* and the synthetic image $\hat{x}$, ensuring that the generated image preserves the clinical severity level of the disease. The second loss function is the perceptual loss, which evaluates the divergence between real and synthetic images in the feature space of a perceptual network. This helps accurately reconstruct both pathological and physiological details in the synthetic images. Our approach employs a pre-trained VGG-19 model as the perceptual network. For a specific layer λ and the VGG feature extraction function $F_{V}^{\lambda }$, we define perceptual loss as
(5)$$\begin{array}{c} L_{\mathrm{percept}}=∥F_{V}^{\lambda }\left(x\right)-F_{V}^{\lambda }\left(\hat{x}\right)∥ \end{array}$$

This loss encourages the generator to create images that are not only visually similar to the real images but also match their structural and textural features at different levels of abstraction.

### 2.4. Preprocessing

Given that each database contained images of different sizes, we standardized all images to a uniform size of 512 × 512 pixels. This resizing ensured that the images remained manageable for processing while preserving critical information. After scaling the images, we extracted their corresponding masks and the images were segmented using the Spacial Attention U-net (SA-Unet) model, as proposed by [28]. This segmentation step ensures that the critical regions, such as blood vessels, are accurately extracted, allowing for more precise lesion generation and classification during the image synthesis process. The preprocessing step was crucial for maintaining consistency across datasets and ensuring accurate lesion extraction and style transfer in the subsequent stages of our method.

## 3. Experimental Setting

In this study, we began by collecting and preprocessing several publicly available medical image datasets relevant to diabetic retinopathy and retinal lesions. The datasets described in Table 1 include the Kaggle Diabetic Retinopathy dataset [25], Indian Diabetic Retinopathy Image Dataset (IDRiD) [29], Retinal-Lesions [30], Fine-Grained Annotated Diabetic Retinopathy (FGADR) [31], and Retinal Fundus MultiDisease Image Dataset (RFMiD) [32]. Each dataset provides a rich collection of retinal images that are essential for training and evaluating our synthetic image generation model.

### 3.1. Kaggle Database

The Kaggle Diabetic Retinopathy Detection dataset [25] was developed for a competition on the Kaggle platform to create models capable of identifying diabetic retinopathy in retinal images. The dataset was provided by the EyePACS platform, and sponsored by California Healthcare. It is one of the most widely used datasets for diabetic retinopathy research, containing a total of 88,702 high-resolution images captured under diverse conditions. The primary purpose of this dataset is to classify diabetic retinopathy severity into five categories: No DR, Mild, Moderate, Severe, and Proliferative DR. For our study, we selected a subset of the dataset using 700 randomly selected images from each class, resulting in a total of 3500 images. From this subset, 1400 images were used for training, and the remaining 2100 images were reserved for testing. We opted to use a smaller subset of the dataset than initially available due to the extended training time required for larger datasets and the inherent class imbalance in the original data, with only 712 images available for the Proliferative DR class.

### 3.2. IDRiD Database

The Indian Diabetic Retinopathy Image Dataset (IDRiD, [29]) is a publicly available database created by the Indian Institute of Technology, Delhi. This dataset contains high resolution retinal images with pixel-level annotations for various lesions associated with diabetic retinopathy. Its primary purpose is to facilitate research in the segmentation and classification of diabetic retinopathy lesions, making it a valuable resource for advancing automated detection and analysis of retinal diseases. The pixel-level annotations provided in this dataset are particularly useful for developing and evaluating algorithms focused on identifying and delineating specific pathological features, such as microaneurysms, hemorrhages, and exudates, which are crucial for accurate diagnosis.

### 3.3. Retinal-Lesions Database

The Retinal-Lesions Database [30] was developed to provide detailed annotations of lesions and severity levels in retinal images. It is commonly used for training models that detect and classify various types of lesions associated with diabetic retinopathy. This dataset contains 1842 selected images sourced from the Kaggle Diabetic Retinopathy Detection database. They have been re-labeled by a panel of 45 ophthalmologists. The re-labeling process categorizes the images into five levels of diabetic retinopathy (DR) and eight distinct lesion classes. The comprehensive nature of this dataset with expert annotations makes it a valuable tool for improving the performance of automated systems in accurately detecting and classifying both the severity of DR and the presence of specific lesions.

### 3.4. FGADR Database

The Fine-Grained Annotated Diabetic Retinopathy (FGADR) database, developed by [31], was created to provide extensive resources for the development and evaluation of machine learning models focused on diabetic retinopathy. This database contains 2842 images, which are divided into two subsets: 1842 images with pixel-level annotations for lesions and 1000 images with lesion grade levels of diabetic retinopathy, assessed by six ophthalmologists. For our work, we utilized the first subset of 1842 images with pixel-level annotations. However, since our focus was on verifying the capacity of lesion transfer in the synthetic images, we opted not to use the annotations in our process. The availability of numerous annotated lesions made this dataset suitable for testing the effectiveness of our lesion transfer techniques in generating medically plausible synthetic images.

### 3.5. RFMID Database

The Retinal Fundus MultiDisease Image Dataset (RFMID) [32] is a comprehensive dataset containing retinal images captured for the diagnosis of multiple retinal diseases. This dataset was designed to address a variety of visual health issues, including diabetic retinopathy, age-related macular degeneration, glaucoma, and several other retinal pathologies. Each image in this database is labeled with high precision, identifying the specific diseases present. We utilized this dataset to demonstrate the versatility of our method, showing its applicability beyond diabetic retinopathy by testing it on various retinal diseases.

### 3.6. Image Metrics

To evaluate the quality of the generated synthetic images, we used several established image metrics that provide quantitative insights into fidelity, diversity, and structural consistency. These metrics ensure a comprehensive assessment of the performance of our proposed WGAN-GP method.

Fréchet Inception Distance (FID) [33]: The FID metric is widely used to measure the similarity between the distributions of real and generated images. It does so by comparing the mean and covariance of features extracted from an Inception [34] network trained on the ImageNet [35] dataset. Lower FID scores indicate that the generated images are closer in distribution to the real images, reflecting higher image quality and realism. This metric is particularly useful in evaluating GANs because it accounts for both the diversity of the generated images (by measuring how well they capture the distribution of the real data) and their fidelity (by assessing how realistic individual images appear). Unlike simple pixel-wise comparisons, FID considers the underlying feature distribution, making it more robust to minor variations in pixel values. Let $\mu_{r},\Sigma_{r}$ be the mean and covariance of the features for the real images and $\mu_{g},\Sigma_{g}$ be the corresponding values for the generated images. The FID score is defined as:
(6)$$\begin{array}{c} \mathrm{FID}=\left|\right|\mu_{r}-\mu_{g}{\left|\right|}^{2}+\mathrm{Tr}\left(\Sigma_{r}+\Sigma_{g}-2{\left(\Sigma_{r}\Sigma_{g}\right)}^{1/2}\right) \end{array}$$Mean Squared Error (MSE): MSE measures the average squared difference between the pixel values of the real and generated images. Lower MSE values correspond to higher image similarity, indicating more accurate reproduction of the original images. Let *r* and *g* represent the real and generated images, respectively, with *N* being the total number of pixels. The MSE formula is defined by
(7)$$\begin{array}{c} MSE=\frac{1}{N}\sum_{i=1}^{N}{\left(x_{i}-y_{i}\right)}^{2} \end{array}$$Structural Similarity Index (SSIM) [36]: SSIM assesses the perceived quality of images by comparing luminance, contrast, and structure between two images. An SSIM value closer to 1 indicates higher structural similarity between the real and synthetic images, capturing key visual information beyond simple pixel-wise differences. Let *r* and *g* be the real and generated images. The SSIM formula is defined as
(8)$$\begin{array}{c} SSIM\left(r,g\right)=\frac{\left(2\mu_{r}\mu_{g}+C_{1}\right)\left(2\sigma_{rg}+C_{2}\right)}{\left(\mu_{r}^{2}+\mu_{g}^{2}+C_{1}\right)\left(\sigma_{r}^{2}+\sigma_{g}^{2}+C_{2}\right)} \end{array}$$
where $\mu_{r}$ and $\mu_{g}$ are the means, $\sigma_{r}^{2}$ and $\sigma_{g}^{2}$ are the variances, $\sigma_{rg}$ is the covariance, and $C_{1}$ and $C_{2}$ are constants for stabilization. SSIM values closer to 1 indicate higher similarity. This metric is particularly relevant for GANs as it assesses perceptual image quality by focusing on structural content rather than individual pixel values.

## 4. Results

We compare our results with PathoGAN from [13] and a cGAN. For each dataset, we generated five synthetic datasets, and the results are summarized in the following tables.

### 4.1. Retinal-Lesions Database

Table 2 presents the results of our experiments with the Retinal-Lesions database. The test set consisted of 1256 images, and we generated an equivalent number of synthetic images based on their corresponding segmented images. The first column lists the different methods we evaluated. Among the configurations tested, we found that the WGAN-GP with nearest-neighbor interpolation consistently outperformed the other methods across all metrics. In Figure 4, we illustrate examples where the original image did not contain lesions. The synthetic images generated by WGAN-GP demonstrate successful synthesis of the overall color, retinal blood vessels, and a well-formed optic disc that is clearly visible. In contrast, the images generated by the cGAN exhibit several issues, such as areas with incorrect coloration, staining, and a poorly formed optic disc. The images generated by PathoGAN are slightly better than those from the cGAN, showing a more consistent structure and an improved optic disc, though still not as refined as those from the WGAN-GP.

### 4.2. FGADR Database

Table 3 presents the results of our evaluations on the FGADR database. This dataset posed significant challenges to the implementation of our method. In this case, the cGAN achieved the best FID score, with a value of 28.16. On the other hand, WGAN-GP with nearest-neighbor interpolation performed best in terms of MSE, achieving a value of 0.544, while WGAN-GP with bilinear interpolation yielded the highest SSIM score, with a value of 0.793.

The difficulty in achieving optimal performance with WGAN-GP on this dataset is likely due to the inherent noise in the images, making it harder for the model to learn effectively. This issue highlights the potential for further improvements when dealing with noisy datasets. Despite the lower FID score, the visual quality of the images generated by WGAN-GP remains comparable to the originals, as shown in Figure 5, where the color and structure of the generated images are similar to the real images.

### 4.3. IDRID Database

For the IDRiD database, the nearest-neighbor configuration proved to be optimal, achieving an FID score of 71.54. The cGAN configuration achieved the lowest MSE, with a value of 0.00722, while the Mitchell-Bicubic configuration attained the highest SSIM, with a score of 0.6852, as shown in Table 4. Due to the small size of this dataset, some features were not captured in full detail, leading to potential overfitting in the model. This issue is evident in the images shown in Figure 6, where the generated images display good color representation and capture key characteristics of retinal images, such as the veins, optic disc, and macula. However, some finer details may have been lost due to the dataset’s limited size.

### 4.4. Kaggle Database

In the Kaggle Diabetic Retinopathy dataset, the nearest-neighbor algorithm yielded the best results, achieving an FID score of 15.21. The bilinear antialiasing configuration performed best in terms of MSE and SSIM, with values of 0.002025 and 0.89, as can be seen in Table 5. This dataset presented a significant challenge in terms of processing time due to its large size, with training times ranging from 3 to 9 days, depending on the configuration. Due to its complexity, the bicubic algorithm took the longest, requiring 9 days of training. It is worth noting that, in the literature, there is only one study that uses a cGAN with this dataset [14], but the code was not made publicly available, which prevented direct comparison with our results.

Figure 7 presents sample images generated with each configuration. It is evident that WGAN-GP successfully synthesizes the key characteristics of retinal images, such as color, blood vessels, and optic disc. In contrast, the cGAN struggled to accurately transfer the images’ color and structural details.

### 4.5. RFMiD

For the RFMiD database, we tested the best overall configuration from previous experiments—namely, the nearest-neighbor method. This dataset was used to assess the capability of the proposed method to generate lesions in medical contexts other than diabetic retinopathy.

In Figure 8, we present a comparison of three real images and three synthetic images generated using the nearest-neighbor configuration. The results, summarized in Table 6, show an FID score of 69.25, comparable to the quality achieved with the IDRiD images. This suggests that the proposed method produces synthetic images of acceptable quality, even when applied to different retinal pathologies. Concerning lesion transfer, Figure 8 shows that, despite the method not being specifically designed for this context, it was able to detect and transfer lesions to the synthetic images successfully.

### 4.6. Lesions

Images containing lesions were also generated, as depicted in Figure 9, Figure 10, Figure 11 and Figure 12 which display examples from all four databases. In these images, the real counterparts feature visible lesions. The results indicate that the proposed method effectively captures and synthesizes the lesions in a manner faithful to the originals. The generated images maintain the overall structure of the retina, including key characteristics such as color, veins, and the optic disc, without introducing artifacts or distortions. In contrast, the images generated by cGAN and PathoGAN do not successfully transfer the lesions, and the colors diverge noticeably from the original images. Additionally, artifacts and a less defined optic disc are present in the synthetic images, except for the IDRiD database, where the small dataset size leads to overfitting, resulting in closer color tones to the originals. Specifically, in Figure 10, although cGAN achieved the best FID score, it does not transfer the lesions as effectively as our improved WGAN-GP method. One limitation of our method is that the lesions appear slightly blurred compared with the real images, but overall, it still outperforms cGAN in lesion synthesis and image quality.

### 4.7. Expert Evaluation

To assess how convincingly the generated images could be perceived as real, a survey was conducted using 50 randomly selected real images from the Retinal-Lesions dataset and 50 synthetic images generated by our method. For each image, respondents were asked to determine whether the image was real or synthetic and rate the overall quality of the image.

The survey was completed by three experts in the field of ophthalmology. Each expert evaluated the images independently, ensuring a diversity of perspectives based on their specific fields of expertise.

Table 7 presents the accuracy of each expert in distinguishing between real and synthetic images. A lower accuracy corresponds to a higher quality of the generated images, as it implies that the experts had difficulty distinguishing between real and synthetic images. Additionally, the table includes the fidelity score, which represents the respondents’ average rating (on a scale from 1 to 10) for each synthetic image. The results indicate that the experts correctly identified only 56.66% of the images (a mix of real and synthetic images generated by the WGAN-GP method). Since the theoretical probability of randomly selecting a synthetic image is 50%, this result suggests that our synthetic images could mimic real images convincingly. The results demonstrate the high quality of the generated images regarding color, structure, and texture, as even trained professionals found it challenging to differentiate between real and synthetic images.

## 5. Discussion

The proposed method based on WGAN-GP with nearest neighbor has demonstrated superior performance across multiple datasets for generating synthetic medical images, particularly for diabetic retinopathy. Compared with existing approaches like cGAN and PathoGAN, WGAN-GP consistently outperformed key metrics such as FID, MSE, and SSIM in the Retinal-Lesions, FGADR, and Kaggle databases. The method’s ability to mitigate mode collapse, a common issue in GANs, significantly contributed to its success in generating diverse and high-quality images.

One of the standout features of this method is its effectiveness in lesion transfer, where the generated images closely replicated the pathological features of the originals, such as color, vessel structure, and the optic disc. Although the method occasionally produced slightly blurred lesions, this issue could be addressed by incorporating detail refinement techniques in future work.

In the expert evaluation, the high average rating for synthetic images and the 56.66% accuracy in distinguishing between real and synthetic images further validated the realism of the generated images. This evaluation suggests that the proposed method can produce clinically relevant synthetic images that are difficult to differentiate from real ones, reinforcing its potential utility in medical image augmentation for training classification models.

The practical applications of this work are significant for addressing the challenges of data scarcity and class imbalance in medical imaging, particularly in fields like ophthalmology. High-quality synthetic retinal images generated by our WGAN-GP method can be used to augment datasets for training deep learning models tasked with detecting conditions such as diabetic retinopathy, hypertension, and age-related macular degeneration. This augmentation can improve the generalization performance of classification models, leading to more reliable diagnostic tools. Furthermore, by integrating multitasking and multimodal learning principles, the synthetic image generation process can be enhanced to support related tasks such as lesion segmentation, disease grading, and patient-specific diagnosis. For example, a system that generates synthetic retinal images conditioned on patient-specific parameters could help in personalized medicine by simulating disease progression or treatment outcomes.

Dataset-specific challenges highlight areas for improvement, such as the FGADR database’s image noise and the IDRiD database’s small size, leading to overfitting. Incorporating more robust preprocessing techniques or exploring advanced data augmentation strategies could improve the model’s generalization in noisy or small datasets.

While the method performed well overall, training time in large datasets like Kaggle remains a concern, with some configurations taking up to nine days. Despite this, the model’s ability to synthesize high-quality images in large datasets suggests that with further optimization, the method could become more time-efficient without sacrificing image quality.

## 6. Conclusions

In this work, we presented a WGAN-GP-based method with nearest-neighbor interpolation for generating high-quality synthetic medical images, specifically targeting diabetic retinopathy and related retinal diseases. The proposed approach demonstrated superior performance compared with cGAN and PathoGAN, particularly in terms of lesion transfer fidelity and overall image quality, as validated by both quantitative metrics and expert evaluations. Key contributions of this work include the accurate replication of lesions and critical retinal structures and the ability to generate clinically relevant synthetic data, which can enhance medical image classification models. The results highlight the potential of WGAN-GP in addressing data scarcity and improving classification accuracy in medical contexts.

However, challenges such as blurring of lesions, handling noisy data, and managing training time in large datasets remain to be solved. Future work will focus on addressing these limitations by incorporating advanced refinement techniques, improved preprocessing, and possibly exploring semi-supervised or hybrid learning approaches. Additionally, asking experts different questions about the pathological components in the images could provide valuable insights into how clinical professionals interpret our images. This feedback could help us understand potential gaps or strengths in the image analysis process and improve the overall accuracy and relevance of our approach. The findings suggest that the proposed WGAN-GP method is a promising tool for synthetic data generation in medical image analysis, with potential applications in clinical and research environments.

## Figures and Tables

**Figure 1 sensors-25-00167-f001:**
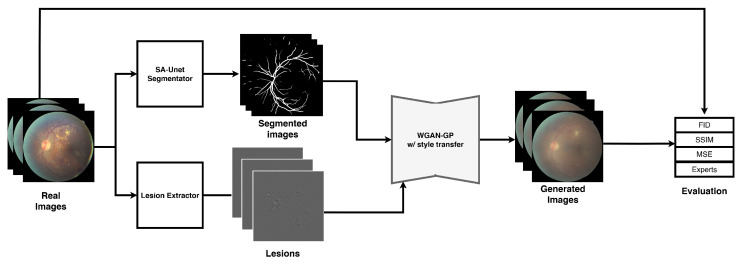
Methodology diagram.

**Figure 2 sensors-25-00167-f002:**
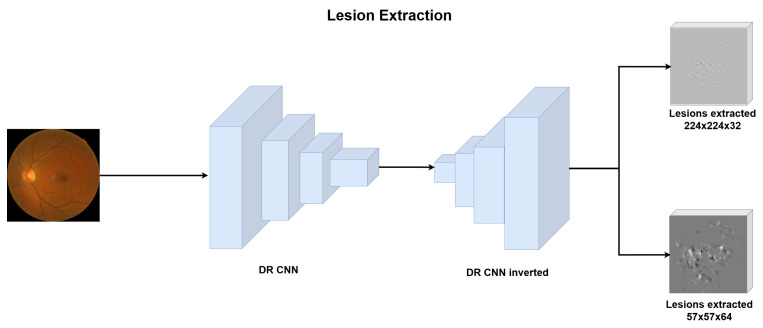
Diagram of the lesion extraction technique.

**Figure 3 sensors-25-00167-f003:**
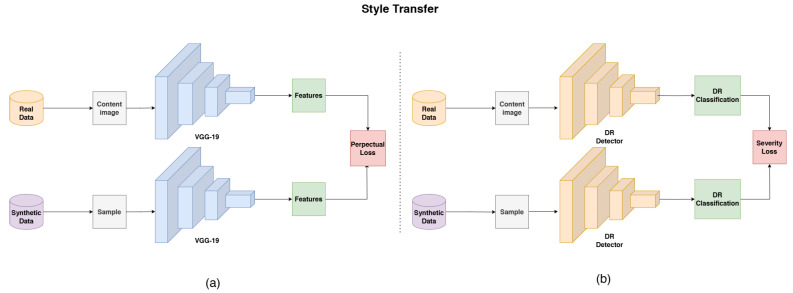
Style Transfer diagrams. (**a**) Diagram illustrating the perceptual loss process, utilizing VGG19 for feature extraction. (**b**) Diagram depicting the severity loss process, where a pretrained CNN is employed for retinal classification.

**Figure 4 sensors-25-00167-f004:**
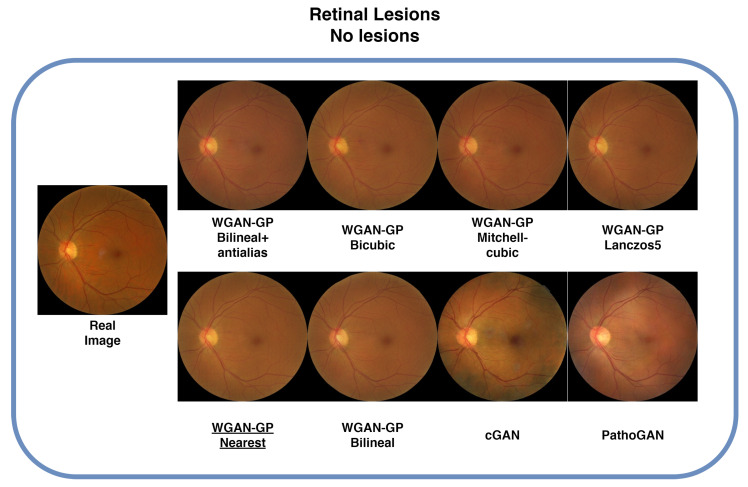
Images of each configuration where the real image does not have lesions. The image with PathoGAN label is the implementation of [13]. The others are using WGAN-GP with different resizing algorithms, except cGAN. Underlined is the best FID result.

**Figure 5 sensors-25-00167-f005:**
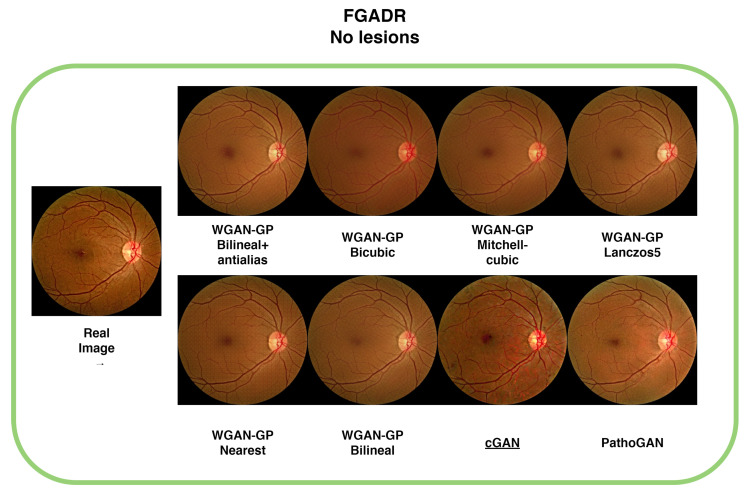
Comparison of generated images across different configurations, where the real image does not contain lesions. The images generated by WGAN-GP and PathoGAN exhibit smoothing effects, while the cGAN successfully transfers the noise from the original image. Underlined is the best FID result.

**Figure 6 sensors-25-00167-f006:**
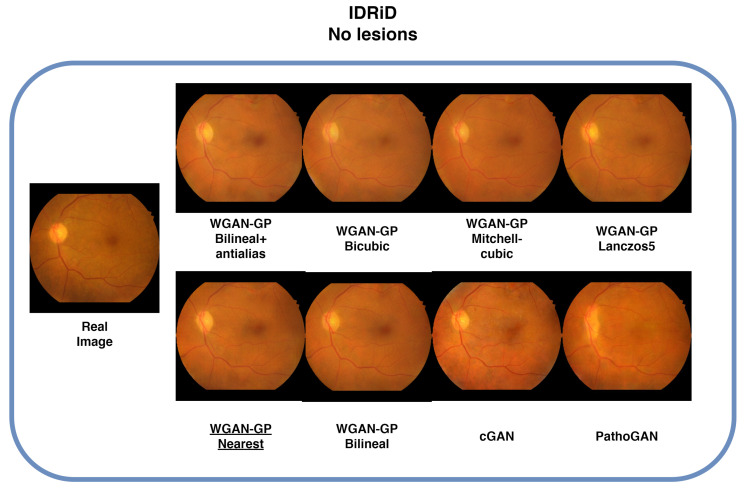
Comparison with generated and real image samples. The images generated using the proposed method exhibit colors and textures that are more similar to the real image. In contrast, the images generated by the cGAN and PathoGAN show color variations in areas where the real image does not present them. Underlined is the best FID result.

**Figure 7 sensors-25-00167-f007:**
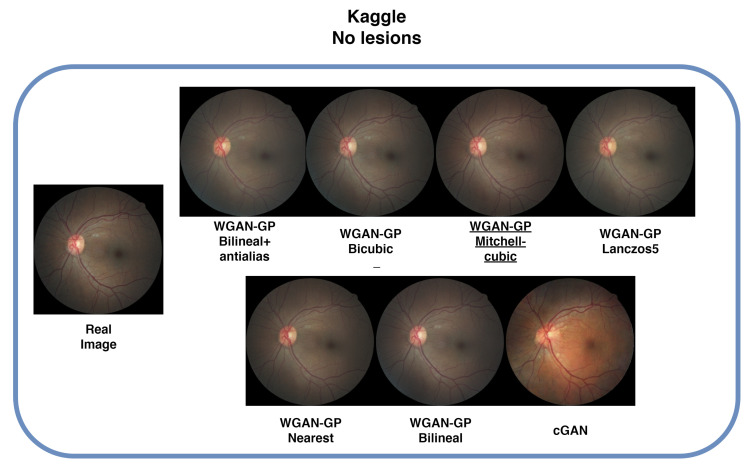
Comparison with generated and real image samples. The proposed method successfully extracts and preserves the color and texture of the original image, while the cGAN method displays different tones. Underlined is the best FID result.

**Figure 8 sensors-25-00167-f008:**
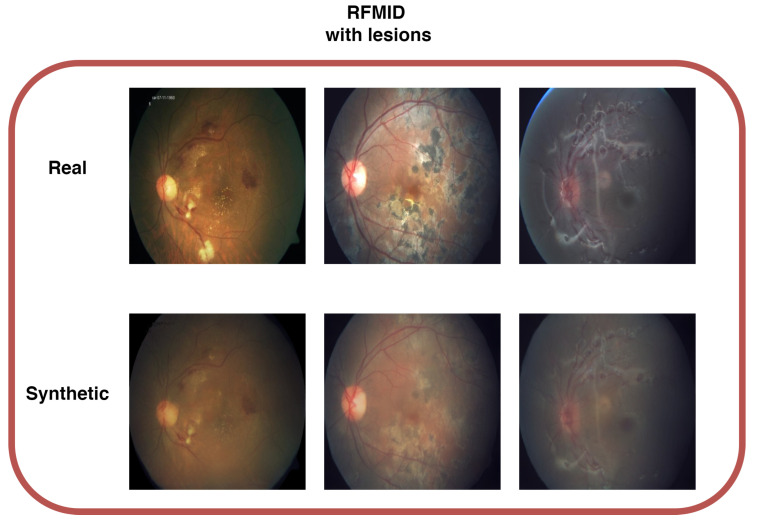
Comparison with generated and real image samples. The proposed method successfully transfers lesions from the original images.

**Figure 9 sensors-25-00167-f009:**
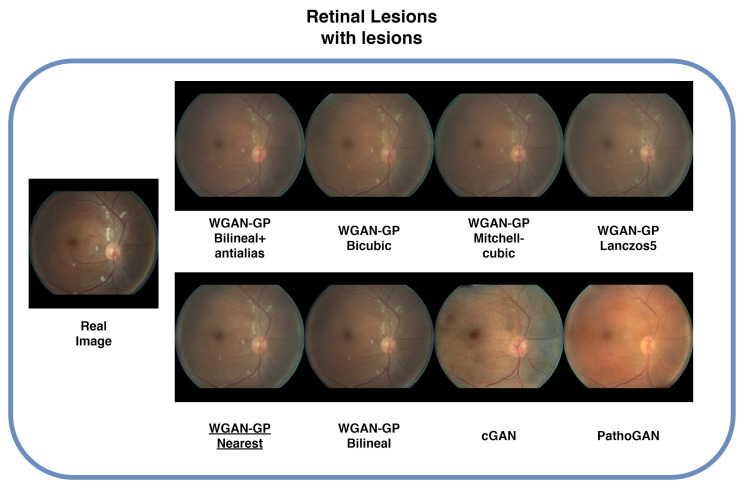
Sample images with lesions from the Retinal-Lesions database. Underlined is the best FID result.

**Figure 10 sensors-25-00167-f010:**
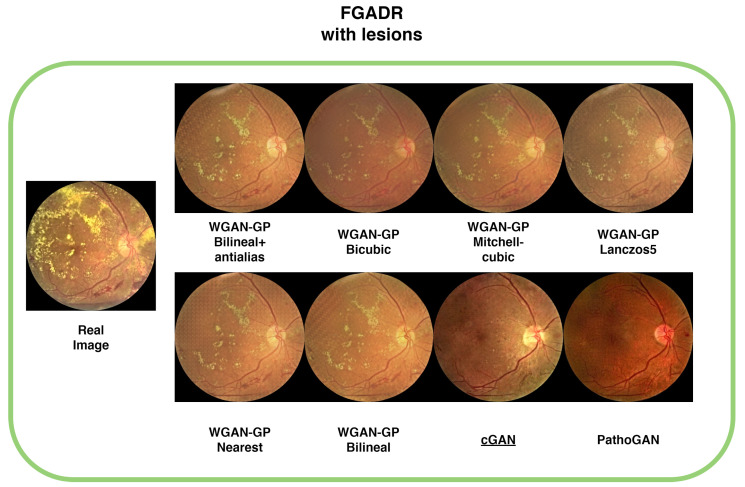
Sample images with lesions from the FGADR database. Underlined is the best FID result.

**Figure 11 sensors-25-00167-f011:**
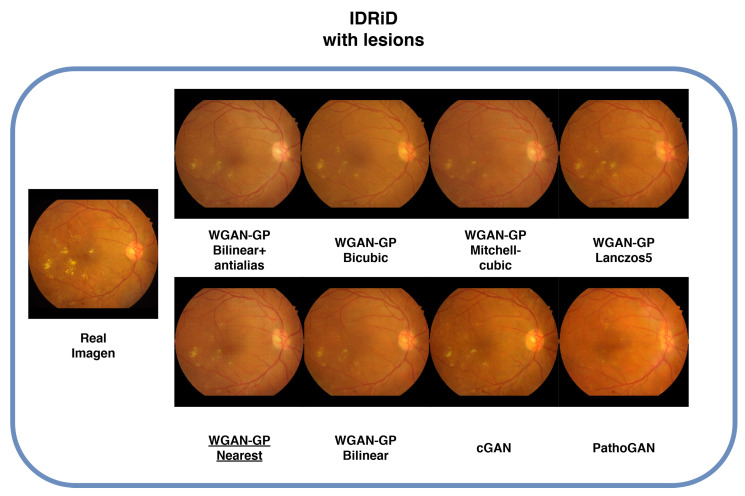
Sample images with lesions from the IDRiD database. Underlined is the best FID result.

**Figure 12 sensors-25-00167-f012:**
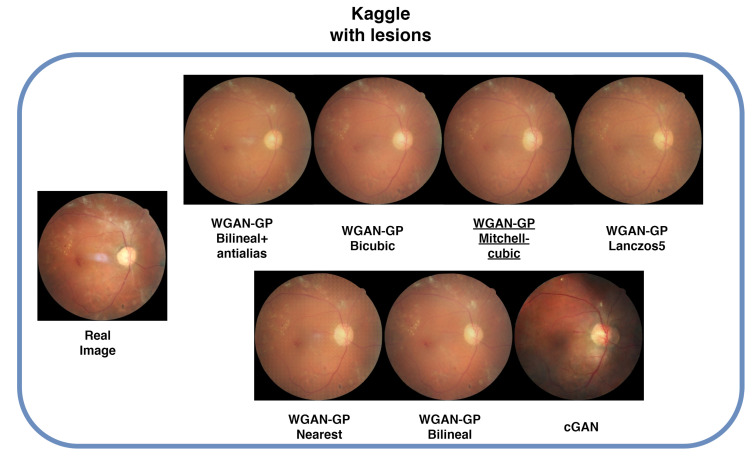
Sample images with lesions from the Kaggle database. Underlined is the best FID result.

**Table 1 sensors-25-00167-t001:** Description of the databases used.

Database	Description	Resolution	Number of Images
Kaggle	DR images, classified in 5 classes	1444 × 1444, 2184 × 3456	1379 train, 2069 test
IDRiD	DR images with segmented lesions	4288 × 2848	54 train, 27 test
Retinal-Lesions	DR images with lesion annotations and severity grading	896 × 896	337 train, 1256 test
FGADR	DR images with lesion segmentation and severity grading	1280 × 1280	500 train, 1342 test
RFMiD	Images with different illness present, labeled by illness	2048 × 1536, 512 × 512	348 train, 174 test

**Table 2 sensors-25-00167-t002:** Experimental results on the Retinal-Lesions database. The arrows (↓, ↑) indicate the desired direction for performance improvement: ↓ denotes that lower values are better (MSE and FID), while ↑ indicates that higher values are preferable (SSIM).

Retinal-Lesions			
**Method**	**MSE** ↓	**SSIM** ↑	**FID** ↓
PathoGAN	0.012045±0.00002	0.8258±0.00004	24.45±0.15
cGAN	0.00902±0.00001	0.8143±0.00003	30.09±0.12
WGAN-GP w/mitchellbicubic	0.00414±0.00001	0.8736±0.00004	21.37±0.03
WGAN-GP w/bicubic	0.00417±0.00001	0.8724±0.00003	20.62±0.06
WGAN-GP w/nearest	0.00358±0.00001	0.8759±0.00005	15.87±0.06
WGAN-GP w/lanczos5	0.00387±0.00001	0.8747±0.00005	20.55±0.05
WGAN-GP w/bilinear	0.00391±0.00001	0.8739±0.00004	18.51±0.03
WGAN-GP w/bilinear+antialias	0.00441±0.00001	0.8717±0.00002	24.56±0.02

**Table 3 sensors-25-00167-t003:** Experimental results on the FGADR database. The arrows (↓, ↑) indicate the desired direction for performance improvement: ↓ denotes that lower values are better (MSE and FID), while ↑ indicates that higher values are preferable (SSIM).

FGADR			
**Method**	**MSE** ↓	**SSIM** ↑	**FID** ↓
PathoGAN	0.010445±0.00004	0.5318±0.00004	24.61±0.13
cGAN	0.01106±0.00002	0.5134±0.00005	21.72±0.10
WGAN-GP w/mitchell-bicubic	0.00802±0.00001	0.5692±0.00003	43.05±0.08
WGAN-GP w/bicubic	0.00863±0.00002	0.5649±0.00008	46.31±0.03
WGAN-GP w/nearest	0.00544±0.00001	0.5692±0.00004	30.61±0.04
WGAN-GP w/lanczos5	0.00582±0.00001	0.5729±0.00006	28.16±0.09
WGAN-GP w/bilinear	0.00569±0.00001	0.5793±0.00004	28.81±0.08
WGAN-GP w/bilinear+antialias	0.00554±0.00001	0.5779±0.00005	31.10±0.03

**Table 4 sensors-25-00167-t004:** Experimental results on the IDRiD database. The arrows (↓, ↑) indicate the desired direction for performance improvement: ↓ denotes that lower values are better (MSE and FID), while ↑ indicates that higher values are preferable (SSIM).

IDRiD			
**Method**	**MSE** ↓	**SSIM** ↑	**FID** ↓
PathoGAN	0.00847±0.0006	0.66024±0.0009	79.88±0.36
cGAN	0.00722±0.00004	0.6576±0.0004	82.88±1.02
WGAN-GP w/mitchellbicubic	0.00764±0.00002	0.6852±0.0003	79.99±0.39
WGAN-GP w/bicubic	0.00751±0.00005	0.6762±0.0002	74.43±0.65
WGAN-GP w/nearest	0.00742±0.00006	0.6699±0.0004	71.54±0.55
WGAN-GP w/lanczos5	0.00727±0.00008	0.6831±0.0005	72.80±0.62
WGAN-GP w/bilinear	0.00729±0.00003	0.6804±0.0005	74.43±0.65
WGAN-GP w/bilinear+antialias	0.00787±0.00007	0.6781±0.0007	72.93±0.70

**Table 5 sensors-25-00167-t005:** Experimental results on the Kaggle database. The arrows (↓, ↑) indicate the desired direction for performance improvement: ↓ denotes that lower values are better (MSE and FID), while ↑ indicates that higher values are preferable (SSIM).

Kaggle			
**Method**	**MSE** ↓	**SSIM** ↑	**FID** ↓
cGAN	0.00902±0.00001	0.8123±0.0006	31.85±0.08
WGAN-GP w/mitchellbicubic	0.00323±0.00006	0.8809±0.0003	15.21±0.56
WGAN-GP w/bicubic	0.00292±0.00008	0.8783±0.0003	16.93±0.68
WGAN-GP w/nearest	0.00350±0.00004	0.8748±0.0005	16.00±0.65
WGAN-GP w/lanczos5	0.00445±0.00008	0.8724±0.0006	19.30±0.55
WGAN-GP w/bilinear	0.00312±0.00005	0.8821±0.0001	16.06±0.45
WGAN-GP w/bilinear+antialias	0.00275±0.00006	0.8850±0.0005	16.85±0.75

**Table 6 sensors-25-00167-t006:** Experimental results on the RFMID dataset. The arrows (↓, ↑) indicate the desired direction for performance improvement: ↓ denotes that lower values are better (MSE and FID), while ↑ indicates that higher values are preferable (SSIM).

RFMiD			
**Method**	**MSE** ↓	**SSIM** ↑	**FID** ↓
WGAN-GP w/nearest	0.0044±0.0001	0.82±0.01	69.25±0.02

**Table 7 sensors-25-00167-t007:** Table of the accuracy and ratings given by the surveyed experts.

Respondents	Accuracy	Rating
Expert 1	52%	8.84
Expert 2	58%	7.06
Expert 3	60%	7.84
Average	56.66%	7.91

## Data Availability

The original data presented in the study are openly available in github.com/hector-anaya.

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
