# Peer review of "WGAN-GP for Synthetic Retinal Image Generation: Enhancing Sensor-Based Medical Imaging for Classification Models"

_sensors, 2024, doi:10.3390/s25010167_

Round 1
Reviewer 1 Report
Comments and Suggestions for Authors
The authors have proposed a technique employing WGAN-GP to generate synthetic retinal images for classification tasks. The method was validated using various publicly available datasets. The following suggestions may be considered to further improve the quality of the paper:
1. Excessive Use of Abbreviations in the Abstract: Simplify the abstract by reducing the number of abbreviations to enhance readability and accessibility.
2. Highlighting Contributions in the Abstract: Clearly emphasize the specific contributions of the study in the abstract to underscore its significance.
3. Enhancing the Introduction: In the introduction (lines 1–68), elaborate on the current challenges in retinal image classification and related diagnostic techniques. This will help highlight the importance of the proposed method and better identify the research gap.
4. Improving Figures: Enhance the quality of figures. For instance, in Figure 1, lesions and other small features in the images are not clearly visible.
5. Refining Expert Feedback Process: Instead of merely asking experts whether an image is synthetic or real, request them to identify specific features that inform their conclusions. This approach could provide valuable insights to further refine the technique in the future.
6. Updating References: Improve the references by citing more recent and relevant works that align with and support the scope of the study.
Reviewer 2 Report
Comments and Suggestions for Authors
This research explores using a Wasserstein Generative Adversarial Network with Gradient Penalty (WGAN-GP) to generate synthetic retinal images to improve diabetic retinopathy classification models. It represents an important contribution to the field of advanced medical imaging. The lack of high-quality, labeled retinal images is a challenge in this field, and the paper introduces a method to create synthetic images that are virtually indistinguishable from real ones.
The proposed architecture's main feature is the use of WGAN-GP as the main element. This helps avoid mode collapse (a common problem in GANs) and generates high-quality, diverse synthetic images. Moreover, lesion extraction and style transfer techniques are integrated, ensuring that the synthetic images retain crucial pathological features, improving the model's ability to learn and generalize. Furthermore, the method's performance is evaluated across multiple public retinal image datasets (Kaggle, IDRiD, Retinal-Lesions, FGADR, RFMID).
The results indicate that the proposed method has competitive performance than state-of-the art approaches looking at the performance of image quality indicators over the test datasets. The synthetic images generated seem to be very similar to real images, made difficult experts to to differentiate synthetic from a real image.
The work is still perfectible since the time production model is considerable, which indicates opportunity areas for optimizing it.
Reviewer 3 Report
Comments and Suggestions for Authors
1.The citation style of the references in this article does not conform to standard practices, and only some of the references have indicated their sources. Additionally, the formulas in the article have not been numbered. It is recommended to correctly cite the references according to standard practices and to indicate the source for each cited reference. Furthermore, formulas should be numbered for clarity.
2.The article mentions the use of the WGAN-GP strategy to improve the generation of medical images, but it does not provide detailed information on the specific implementation and advantages of this innovation, which may require additional reading of other literature for readers who are unfamiliar with this area to understand the novelty. It is recommended to provide a more detailed description of the implementation method and advantages of the innovation.
3.The article conducts validation on multiple databases, but there may be differences in image quality and lesion types among these databases. Furthermore, when comparing different methods, the article primarily relies on image quality assessment metrics, which may not fully reflect the actual effectiveness of the synthetic images in clinical applications. It is recommended to use multiple databases and incorporate clinical indicators to enhance the reliability of the experiments.
4.The article primarily focuses on the field of medical image generation but fails to provide a comprehensive overview of the current research status and development trends in this area. Consequently, readers may not understand the necessity of the innovation presented. It is recommended to mention the main research directions, existing challenges, and opportunities in the current field of medical image generation, so that readers can better understand the research background and prospects of this area.
5. Synthetic Retinal Image Generation needs to draw on the latest research trends in machine learning and should at least be mentioned in the analysis, multitasking and multimodal machine learning algorithms. Some machine learning algorithms based on multitasking and multimodality have good reference value for this task, such as A lightweight convolutional neural network for road surface classification under shadow interference . , Multi-task learning for hand heat trace time estimation and identity recognition, Deep soft threshold feature separation network for infrared handprint identity recognition and time estimation. Meanwhile, the author should more effectively describe the practical application scope and significance of this article.
6. In making comparisons, the article primarily focuses on traditional GANs and other existing image generation methods, but it does not mention comparisons with the latest research findings. This omission may prevent readers from understanding the position and contributions of the WGAN-GP method within the current research context. It is recommended to include comparisons with the latest research findings to enhance the cutting-edge nature of the article.
Comments on the Quality of English LanguageThe English could be improved to more clearly express the research.
